# Infective Endocarditis in Diabetic Patients: A Different Profile with Prognostic Consequences

**DOI:** 10.3390/jcm11092651

**Published:** 2022-05-09

**Authors:** María Isabel Biezma, Patricia Muñoz, Sofía De la Villa, Mª Carmen Fariñas-Álvarez, Francisco Arnáiz de las Revillas, Encarnación Gutierrez-Carretero, Arístides De Alarcón, Raquel Rodríguez-García, Jaume Llopis, Miguel Ángel Goenaga, Andrea Gutierrez-Villanueva, Antonio Plata, Laura Vidal, Manuel Martínez-Sellés

**Affiliations:** 1Escuela de Doctorado, Universidad Europea de Madrid, 28670 Madrid, Spain; mariabel.biezma@gmail.com; 2Servicio de Microbiología Clínica y Enfermedades Infecciosas, Hospital General Universitario Gregorio Marañón, Instituto de Investigación Sanitaria Gregorio Marañón, 28007 Madrid, Spain; 3CIBERES (CIBER Enfermedades Respiratorias)—Facultad de Medicina, Universidad Complutense de Madrid, 28040 Madrid, Spain; 4Servicio de Enfermedades Infecciosas, Hospital Universitario Marqués de Valdecilla, IDIVAL (Instituto de Investigación Sanitaria Valdecilla), CIBER de Enfermedades Infecciosas-CIBERINFEC (CB21/13/00068), Instituto de Salud Carlos III, Universidad de Cantabria, 39008 Santander, Spain; 5Cardiac Surgery Service, CIBERCV (CIBER Enfermedades Cardiovasculares), Institute of Biomedicine of Seville (IBiS), University of Seville/CSIC/University Hospital Virgen del Rocío Seville, 41013 Sevilla, Spain; 6Servicio de Medicina Intensiva, Hospital Universitario Central de Asturias, Universidad de Oviedo, 33011 Oviedo, Spain; 7Department of Genetics, Microbiology and Statistics, University of Barcelona, 08007 Barcelona, Spain; 8Servicio de Enfermedades Infecciosas, Hospital Universitario Donosti, 20014 San Sebastian, Spain; 9Unidad de Enfermedades Infecciosas, Servicio de Medicina Interna, Universitario Puerta de Hierro, 28222 Majadahonda, Spain; 10UGC Enfermedades Infecciosas, Microbiología y Medicina Preventiva, IBIMA (Instituto de Investigación Biomédica de Málaga), Hospital Regional Universitario de Málaga, 29010 Malaga, Spain; 11Servicio de Cardiología, Hospital Universitario Son Espases, 07120 Palma de Mallorca, Spain; 12Cardiology Department, Hospital General Universitario Gregorio Marañón, CIBERCV (CIBER Enfermedades Cardiovasculares), Universidad Complutense, 28040 Madrid, Spain

**Keywords:** infective endocarditis, diabetes mellitus, prognosis, in-hospital mortality, one-year mortality, organ damage

## Abstract

Background. Infective Endocarditis (IE) is a severe condition. Diabetes mellitus (DM) has been associated with a poor prognosis in other settings. Our aim was to describe the profile and prognosis of IE with and without DM and to analyze the prognostic relevance of DM-related organ damage. Methods. Retrospective analysis of the Spanish IE Registry (2008–2020). Results. The cohort comprises 5590 IE patients with a mean age of 65.0 ± 15.5 years; 3764 (67.3%) were male. DM was found in 1625 patients (29.1%) and 515 presented DM-related organ damage. DM prevalence during the first half of the study period was 27.6% vs. 30.6% in the last half, *p* = 0.015. Patients with DM presented higher in-hospital mortality than those without DM (521 [32.1%] vs. 924 [23.3%], *p* < 0.001) and higher one-year mortality (640 [39.4%] vs. 1131 [28.5%], *p* < 0.001). Among DM patients, organ damage was associated with higher in-hospital (200 [38.8%] vs. 321 [28.9%], *p* < 0.001) and one-year mortality (247 [48.0%] vs. 393 [35.4%], *p* < 0.001). Multivariate analyses showed an independent association of DM with in-hospital (odds ratio [OR] = 1.34, 95% confidence interval [CI]: 1.16–1.55, *p* < 0.001) and one-year mortality (OR = 1.38, 95% CI: 1.21–1.59, *p* < 0.001). Among DM patients, organ damage was independently associated with higher in-hospital (OR = 1.37, 95% CI: 1.06–1.76, *p* = 0.015) and one-year mortality (OR = 1.59, 95% CI = 1.26–2.01, *p* < 0.001) Conclusions. The prevalence of DM among patients with IE is increasing and is already above 30%. DM is independently associated with a poor prognosis, particularly in the case of DM with organ damage.

## 1. Introduction

Infective Endocarditis (IE) is a severe disease with high in-hospital mortality [1,2]. Almost one third of IE patients present diabetes mellitus (DM) [2,3,4,5]. DM has been associated with a poor prognosis in sepsis [6,7,8]. An association of DM with prognosis in IE patients has also been described [1,4]. However, DM is associated with advanced age, comorbidities, atypical clinical presentation, and longer IE diagnosis time, among other characteristics that have a strong prognostic influence in IE [2,3,4,5]. Due to that reason, the independent association of DM with IE mortality is unclear [3,4]. Some authors have suggested an independent association [1] while other data do not support it [3].

Our aim was to describe the profile and prognosis of IE with and without DM and to analyze the prognostic relevance of DM-related organ damage. We also studied the evolution of the yearly prevalence of DM in these patients.

## 2. Methods

The Spanish Collaboration on Endocarditis—GAMES (*Grupo de Apoyo al Manejo de la Endocarditis Infecciosa en España*)—is a national observational registry that has been previously described [9,10,11]. Multidisciplinary teams that compose this group, including infectious disease physicians, cardiologists, cardiac surgeons, microbiologists, echocardiographers, and other imaging specialists, prospectively completed standardized case report forms with information regarding IE episodes and follow-up data. A complete list of GAMES members is shown in Acknowledgments. IE patients were consecutively included at 38 Spanish hospitals between January 2008 and December 2020. Inclusion criteria were the diagnosis of definite or probable IE by modified Duke criteria [12]. IE management, including the decision to perform surgery and the type of surgery, was done by the local medical team following the 2009 and 2015 European Society of Cardiology recommendations [13]. DM was diagnosed based on the American Diabetes Association criteria [14]. DM-associated organ damage was considered to be present after analyzing clinical and laboratory techniques, as well as image data. For instance, renal disease with albuminuria and/or reduced glomerular filtration rate in the absence of signs or symptoms of other primary causes of kidney damage, neuropathy with loss of protective sensation, and neovascularization and/or vitreous/preretinal bleeding (in addition to non-proliferative retinopathy) [15].

This study complies with the principles outlined in the Declaration of Helsinki and was approved by the ethics committee of participating centers.

### Statistical Methods

Continuous variables are summarized as means ± standard deviations (SD) or medians, and interquartile ranges, when a normal distribution was not observed, as per the Kolmogorov–Smirnov goodness-of-fit test; categorical variables are expressed as numbers and percentages. Student‘s *t*-test, Mann–Whitney U test, or paired *t*-test were used to compare continuous variables. Categorical variables were compared using the χ^2^ test or Fisher’s exact test. Multivariable logistic regression analyses (backward selection) were performed to determine mortality predictors and to assess the independent association of DM, with and without organ damage, with mortality. All variables with *p* value < 0.10 in univariate analyses were included in the multivariable analyses. Statistical analysis was performed using SPSS, version 22.0 (IBM, Armonk, NY, USA).

## 3. Results

The cohort comprises 5590 IE patients with a mean age of 65.0 ± 15.5 years; 3764 (67.3%) were male. DM was found in 1625 patients (29.1%) and 515 presented DM-related organ damage (Figure 1).

Figure 2 shows the yearly prevalence of DM during the study period. The prevalence of DM during the first half of the study period was 27.6% vs. 30.6% in the last half, *p* = 0.015.

Table 1 shows the clinical characteristics of patients with and without DM. Compared with those without DM, DM patients presented more frequently advanced age, cardiac implantable electronic device location, nosocomial origin, cardiovascular and renal disease, and had a higher Charlson Comorbidity index. Mean age on the first 6.5 years of the study period was lower than in the last 6.5 years, both in the global population and in diabetic patients (65.0 ± 15.6 years vs. 66.2 ± 15.3 years, *p* = 0.004; 69.3 ± 11.2 vs. 71.0 ± 10.5, *p* = 0.003, respectively).

Staphylococcus and enterococcus etiology were more common among diabetics (Table 2). Table 3 shows the clinical outcome according to the presence of diabetes. Compared with those without DM, DM patients presented complications more frequently and had higher in-hospital and one-year mortality. Among diabetics, patients with DM-related organ damage were a high-risk population with a poor prognosis (Table 4 and Figure 3). Multivariate analyses showed an independent association of DM with in-hospital and one-year mortality (Table 5). In addition, among diabetics, organ damage was an independent predictor of mortality.

## 4. Discussion

Our data show that the prevalence of DM among patients with IE is increasing and about 30% of patients with IE have DM. Diabetics had a poor prognosis, particularly in the case of DM with organ damage. Compared with non-diabetics, diabetic patients had comorbidities more frequently, mainly cardiovascular and renal disease. Diabetics also had a high-risk profile with more nosocomial and healthcare-related IE and more frequent *S. aureus* etiology. As expected, diabetics had a poor prognosis. Even after correcting for confounding factors, the association of DM with in-hospital and one-year mortality remained significant.

The prevalence of DM in the general population is increasing [16]. In our country the prevalence of DM in the general population aged 65–75 years increased from 17% in 2006 to 21% in 2017 [17]. Our study also found a similar trend in IE patients. Previous authors have suggested an increase in DM prevalence in IE patients [3,5]. Abe et al. [5] found a prevalence of 22% in 2004 that increased to 30% in 2014. The reasons that explain the increase in the prevalence of DM are unknown. Population aging may play a role. In our sample, the mean age on the last half of the study period was higher than in the first half, and this was also true in diabetic patients.

Although some previous studies suggested an association of DM with IE mortality [1,4], others did not identify DM as a prognostic factor [18,19,20,21,22,23,24]. The independent association of DM with prognosis is unclear and a significant age interaction could be a confounding factor [4]. On the other hand, even prediabetes has been associated with a higher mortality risk [25,26]. Although no previous studies have focused on the prognostic implications of DM-related organ damage, DM has been associated with higher rates of heart and renal failures [26] and more advanced DM stages, such as longer DM duration, insulin-treated DM [25], and higher Diabetes Complications Severity Index [27], which have been related to higher IE risk and a poor prognosis.

The relation of DM with the prognosis of IE might have several explanations. DM is associated with endothelial dysfunction which can promote stronger bacterial adhesion [28,29]. In addition, diabetics have an impaired immune response [30] and more common bacteremia with aggressive bacteria such as *S. aureus* [31]. Moreover, diabetics have defects of neutrophil activities [32,33,34]. Immune system dysfunction due to chronic low-grade inflammation seen in DM favors micro-organism growth, a process that contributes to sepsis progression [6,7,8].

Our work could have relevant clinical implications. Due to the poor prognosis of IE in diabetics, it might be reasonable to consider earlier and more aggressive treatments and interventions in these patients, particularly in those with previous DM-related organ damage. Close follow-up and correct glycaemia control might improve the outcome.

The limitations of this study should be noted. The retrospective design justifies that relevant variables such as type of DM, DM duration, level of glycated hemoglobin, presence of diabetic cardiomyopathy, and DM-therapy were not collected systematically. Local medical teams were responsible for IE management, including deciding on surgery, and any judgements may have been influenced by factors not registered in this study. Finally, cause of death during follow-up was not available for a large number of patients.

In any case, our data come from a large national database and show a clear association of DM with IE prognosis. Moreover, ours is the first study to compare the prognosis of IE in diabetics with and without organ damage.

## 5. Conclusions

The prevalence of DM among patients with IE is increasing and is already above 30%. DM is independently associated with a poor prognosis, particularly in the case of DM with organ damage.

## Figures and Tables

**Figure 1 jcm-11-02651-f001:**
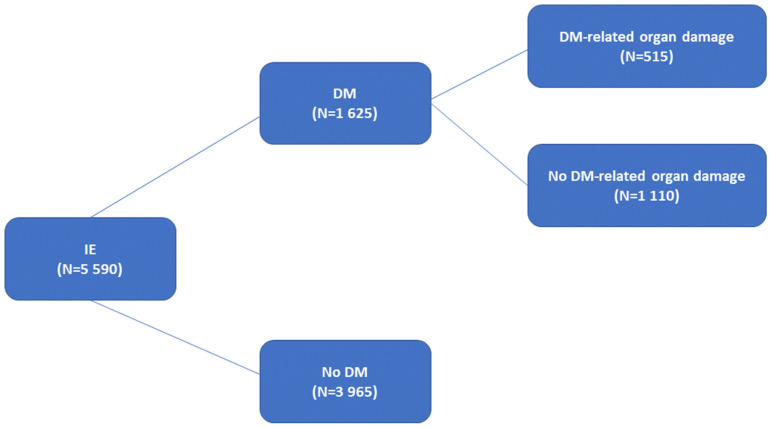
Study overview. IE: infective endocarditis. DM: diabetes mellitus.

**Figure 2 jcm-11-02651-f002:**
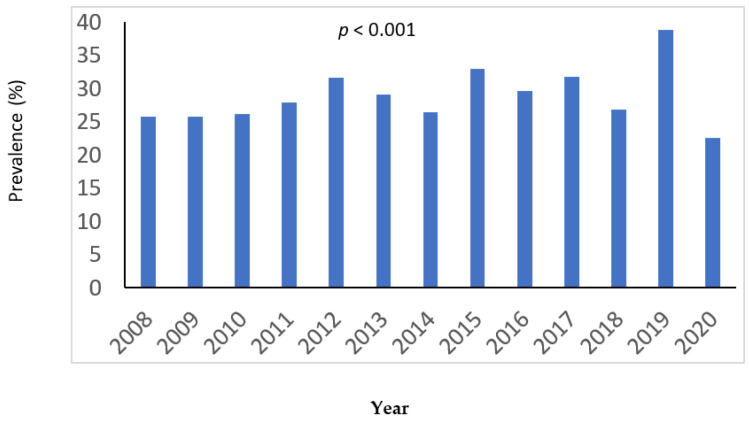
Yearly prevalence of diabetes during the study period.

**Figure 3 jcm-11-02651-f003:**
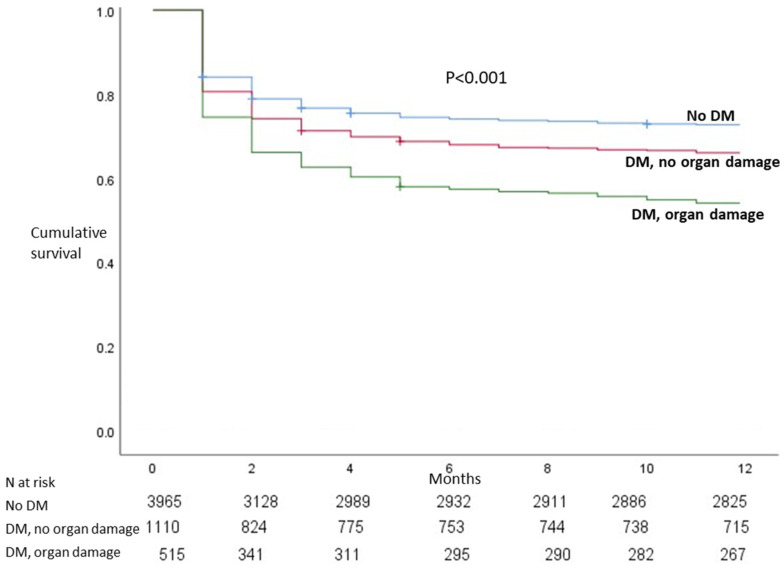
Kaplan–Meier survival according to the presence of diabetes mellitus (DM) and DM-related organ damage.

**Table 1 jcm-11-02651-t001:** Clinical characteristics of infective endocarditis (IE) according to the presence of diabetes mellitus (DM).

	No DM (3965)	DM (1625)	*p*
Age, years, median (IQR)	67 (54–77)	72 (64–78)	<0.001
Male Gender (%)	2679 (67.5)	1085 (66.7)	0.564
Nosocomial IE (%)	1041 (26.3)	535 (32.9)	<0.001
Health care-associated IE (%)	306 (7.7)	158 (9.7)	0.014
Vegetation present (%)	2893 (73.0)	1150 (70.8)	0.096
New heart murmur (%)	1349 (34.0)	477 (29.3)	0.001
**Location (%)**			
Aortic	2060 (52.0)	830 (51.1)	0.551
Mitral	1688 (42.6)	670 (41.2)	0.356
Tricuspid	215 (5.4)	79 (4.9)	0.394
Pulmonary	71 (1.8)	8 (0.5)	<0.001
PM/ICD	349 (8.8)	213 (13.1)	<0.001
Others	113 (2.8)	37 (2.3)	0.229
Multiple	570 (14.4)	227 (14.0)	0.693
Unknown	66 (1.7)	24 (1.5)	0.613
Native IE	2459 (62.0)	941 (57.9)	0.004
Prosthetic IE	1213 (30.6)	528 (32.5)	0.164
**Comorbidities (%)**			
Respiratory disease	668 (16.8)	379 (23.3)	<0.001
Coronary artery disease	877 (22.1)	625 (38.4)	<0.001
Heart failure	1179 (29.7)	691 (42.5)	<0.001
Peripheral arterial disease	286 (7.2)	315 (19.3)	<0.001
Stroke	441 (11.1)	277 (17.0)	<0.001
Cancer	642 (16.1)	254 (15.6)	0.604
Renal disease	844 (21.3)	586 (36.1)	<0.001
Liver disease	370 (9.4)	163 (10.1)	0.443
Congenital heart disease	303 (7.6)	38 (2.3)	<0.001
Native heart valve disease	1794 (45.2)	773 (47.5)	0.113
Age-adjusted Charlson score, median (IQR)	4 (2–6)	6 (5–8)	<0.001

IQR = Interquartile range; PM = Pacemaker; ICD = Implantable cardioverter defibrillator.

**Table 2 jcm-11-02651-t002:** Infective endocarditis etiology according to the presence of diabetes mellitus (DM).

Etiology (%)	No DM (3965)	DM (1625)	*p*
*Staphylococcus aureus*	836 (21.1)	413 (25.4)	<0.001
*Coagulase-negative staphylococcus*	680 (17.2)	322 (19.8)	0.018
*Enterococcus*	519 (13.1)	268 (16.5)	0.001
*Streptococcus*	1081 (27.3)	346 (21.3)	<0.001
*Candida*	68 (1.7)	20 (1.2)	0.187
Other Fungi	11 (0.3)	4 (0.2)	0.837
Unknown	345 (8.7)	112 (6.9)	0.025
Anaerobe	52 (1.3)	16 (1.0)	0.311
Polymicrobial	65 (1.6)	21 (1.3)	0.338
Gram-negative bacteria	167 (4.2)	60 (3.7)	0.372
Other etiologies	112 (2.8)	34 (2.1)	0.119

**Table 3 jcm-11-02651-t003:** Clinical Course according to the presence of diabetes mellitus (DM).

	No DM (3965)	DM (1625)	*p*
**Intracardiac complications (%)**	1264 (31.9)	500 (30.8)	0.715
Valve perforation	581 (14.6)	203 (12.4)	0.035
Pseudoaneurysm	248 (6.2)	90 (5.5)	0.308
Abscess	639 (16.1)	283 (17.4)	0.235
Fistula	105 (2.6)	35 (2.1)	0.283
Vascular events (%)	340 (8.5)	98 (6.0)	0.001
Heart Failure (%)	1554 (39.1)	699 (43.0)	0.008
Persistent bacteremia (%)	402 (10.1)	204 (12.5)	0.008
Central nervous system involvement (%)	750 (18.1)	350 (21.5)	0.025
Embolism (%)	888 (22.3)	316 (19.4)	0.015
Renal Failure, new or worsening (%)	1341 (33.8)	654 (40.2)	<0.001
Sepsis (%)	632 (15.9)	300 (18.4)	0.022
Septic shock (%)	468 (11.8)	226 (13.9)	0.03
Cardiac Surgery (%)	1905 (48.0)	707 (43.5)	0.002
In-hospital mortality (%)	924 (23.3)	521 (32.1)	<0.001
One-year mortality (%)	1131 (28.5)	640 (39.4)	<0.001

**Table 4 jcm-11-02651-t004:** Main differences seen in patients with diabetes according to the presence of diabetes-related organ damage.

	No Organ Damage (1110)	Organ Damage (515)	*p*
Nosocomial IE (%)	330 (29.7)	205 (39.8)	<0.001
Health care-associated IE (%)	67 (6.0)	91 (17.7)	<0.001
Native IE (%)	617 (55.6)	324 (62.9)	0.005
Prosthetic IE (%)	392 (35.3)	136 (26.4)	<0.001
Coronary artery disease (%)	394 (35.4)	231 (44.8)	<0.001
Heart failure (%)	428 (38.5)	263 (51.0)	<0.001
Peripheral arterial disease (%)	116 (10.4)	199 (38.4)	<0.001
Stroke (%)	168 (15.1)	109 (21.1)	0.003
Cancer (%)	197 (17.7)	57 (11.0)	0.001
Renal disease (%)	267 (24.1)	319 (61.9)	<0.001
Age-adjusted Charlson score, median (IQR)	6 (4–7)	8 (6–9)	<0.001
*Staphylococcus aureus* (%)	219 (19.7)	194 (37.7)	<0.001
*Streptococcus* (%)	278 (25.0)	68 (13.2)	<0.001
Cardiac Surgery (%)	523 (47.1)	184 (35.7)	<0.001
In-hospital mortality (%)	321 (28.9)	200 (38.8)	<0.001
One-year mortality (%)	393 (35.4)	247 (48.0)	<0.001

IQR = Interquartile range.

**Table 5 jcm-11-02651-t005:** Independent predictor of mortality. (**A**) All population, in-hospital mortality. (**B**) All population, one-year mortality. (**C**) Diabetics, in-hospital mortality. (**D**) Diabetics, one-year mortality.

**(A)**		
	**OR (95% CI)**	** *p* **
Diabetes	1.4 (1.2–1.6)	<0.001
Age (years)	1.02 (1.02–1.03)	<0.001
Female sex	1.3 (1.1–1.5)	0.001
Heart failure	2.6 (2.3–3.0)	<0.001
Renal disease	2.2 (1.9–2.5)	<0.001
Sepsis	2.1 (1.7–2.5)	0.005
*S. aureus*	1.4 (1.2–1.7)	0.002
**(B)**		
	**OR (95% CI)**	** *p* **
Diabetes	1.4 (1.2–1.6)	<0.001
Age (years)	1.02 (1.02–1.03)	<0.001
Female sex	1.17 (1.02–1.34)	0.027
Heart failure	2.4 (2.1–2.8)	<0.001
Renal disease	2.0 (1.8–2.3)	<0.001
Sepsis	2.0 (1.7–2.4)	<0.001
Cardiac surgery not done	1.3 (1.1–1.5)	0.005
*S. aureus*	1.3 (1.1–1.5)	0.002
**(C)**		
	**OR (95% CI)**	** *p* **
Diabetes-related organ damage	1.4 (1.1–1.8)	0.01
Age (years)	1.02 (1.01–1.03)	0.007
Female sex	1.4 (1.1–1.8)	0.01
Heart failure	3.2 (2.5–4.1)	<0.001
Renal disease	2.3 (1.8–2.9)	<0.001
Sepsis	1.8 (1.3–2.4)	<0.001
*S. aureus*	1.4 (1.1–1.9)	0.01
**(D)**		
	**OR (95% CI)**	** *p* **
Diabetes-related organ damage	1.6 (1.3–2.0)	<0.001
Age (years)	1.02 (1.01–1.03)	<0.001
Female sex	1.32 (1.04–1.67)	0.024
Heart failure	2.7 (2.1–3.3)	<0.001
Renal disease	2.1 (1.7–2.6)	<0.001
Sepsis	1.7 (1.3–2.3)	<0.001

OR = Odds ratio; CI = Confidence Interval.

## Data Availability

Data available under request from SEICAV.

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
