# Peer review of "Infective Endocarditis in Diabetic Patients: A Different Profile with Prognostic Consequences"

_jcm, 2022, doi:10.3390/jcm11092651_

Round 1

Reviewer 1 Report

The paper entitled: “Infectiveendocarditisin diabeticpatients: adifferent profile withprognosticconsequences” presented to reviews an important analysis of the cohort comprises 5.590 patients with infective endocarditis (IE) in respect to in-hospital and one-year mortality. In 1.625 patients was found diabetes mellitus (DM). In the group of diabetic patients - 515 presented DM-related organ damage. At this point, I would like to mention that the Authors did not select group of patients with diabetic cardiomyopathyas a clinical condition of myocardial dysfunction that occurs in the absence of coronary atherosclerosis and hypertension in patients with diabetes mellitus. Diabetic cardiomyopathyis a significant contributor to the high mortality rates associated with diabetes.

            The Authors have shown conclusively the association of diabetes mellitus particularly in the case of DM with organ damage with both in-hospital and one-year mortality. At this point, I would like to point out that only the overall mortality rate was assessed. The authors do not give a cause of death (cardiac death or cardiac arrest and noncardiac death). Deaths from cardiac causes appear to be of particular clinical importance and it should be carefully analyzed by Authors.

At the same time, I would like to point out some minor shortcomings:

  1. No description of the axis of the plot located in figure 2.
  2. The reference list should be formatted according to the journal style.

I pay particular attention to the number of authors of the cited publications, e.g. literature number 32 - there are 3 authors and et al. ; while in many other literature items, e.g. literature number 23 - there are as many as 12 authors. It should be harmonized according to the JCM guidelines.

Author Response

We would like to thank the comments from Reviewer 1 that have helped us to improve our manuscript.

The paper entitled: “Infective endocarditis in diabetic patients: a different profile with prognostic consequences” presented to reviews an important analysis of the cohort comprises 5.590 patients with infective endocarditis (IE) in respect to in-hospital and one-year mortality. In 1.625 patients was found diabetes mellitus (DM). In the group of diabetic patients - 515 presented DM-related organ damage.

Thank you for this comment.

At this point, I would like to mention that the Authors did not select group of patients with diabetic cardiomyopathy as a clinical condition of myocardial dysfunction that occurs in the absence of coronary atherosclerosis and hypertension in patients with diabetes mellitus. Diabetic cardiomyopathy is a significant contributor to the high mortality rates associated with diabetes.

Unfortunately, the variable “diabetic cardiomyopathy” was not included in GAMES database, we have included this as a limitation.

            The Authors have shown conclusively the association of diabetes mellitus particularly in the case of DM with organ damage with both in-hospital and one-year mortality. At this point, I would like to point out that only the overall mortality rate was assessed. The authors do not give a cause of death (cardiac death or cardiac arrest and noncardiac death). Deaths from cardiac causes appear to be of particular clinical importance and it should be carefully analyzed by Authors.

Again, cause of death is not available in GAMES follow-up, please note most deaths are obtained from administrative data and the exact cause of death is difficult to know. This has also been included as a limitation.

At the same time, I would like to point out some minor shortcomings:

  1. No description of the axis of the plot located in figure 2.

Sorry for this mistake that has been corrected.

  1. The reference list should be formatted according to the journal style.

I pay particular attention to the number of authors of the cited publications, e.g. literature number 32 - there are 3 authors and et al. ; while in many other literature items, e.g. literature number 23 - there are as many as 12 authors. It should be harmonized according to the JCM guidelines.

This has been done.

Reviewer 2 Report

Dr. Biezma et al. retrospectively analyze the profile and prognosis of infective endocarditis with and without diabetes. They conclude that the prevalence of DM among patients with IE is increasing. DM is independently associated with a poor prognosis, particularly in the case of DM with organ damage.

The manuscript is well-organized and the results are interesting.

Major comments:

  1. Were the duration of DM and HbA1c at admission associated with the prognosis?
  2. Non-DM patients had more cardiac surgery, compared with DM patients. Was that because the severity of the disease (it seemed that non-DM patients had more valve perforation and new heart murmur)? Was the cardiac surgery associated with the mortality?
  3. What was the age during the first half of the study period vs the last half?

Author Response

We would like to thank the comments from Reviewer 2 that have helped us to improve our manuscript.

Dr. Biezma et al. retrospectively analyze the profile and prognosis of infective endocarditis with and without diabetes. They conclude that the prevalence of DM among patients with IE is increasing. DM is independently associated with a poor prognosis, particularly in the case of DM with organ damage.

 The manuscript is well-organized and the results are interesting.

Thank you for this comment.

  1. Were the duration of DM and HbA1c at admission associated with the prognosis?

As the reviewer points out, this is a retrospective analysis and duration of DM and HbA1c at admission are not included in GAMES database, this is why we included it as a limitation. In any case, the presence of diabetes-related organ damage is probably correlated with the duration of DM and HbA1C levels.

  1. Non-DM patients had more cardiac surgery, compared with DM patients. Was that because the severity of the disease (it seemed that non-DM patients had more valve perforation and new heart murmur)? Was the cardiac surgery associated with the mortality?

As we explain in methods and include in limitations, IE management, including the decision to perform surgery and type of surgery, was done by the local medical team. Due to this reason, we cannot be certain of the influence of the severity of disease on the decision to perform surgery.

As can be seen in table 5B, cardiac surgery was associated with one-year survival.

  1. What was the age during the first half of the study period vs the last half?

Mean age on the first 6.5 years of the study period was lower than in the last 6.5 years, both in the global population and in diabetic patients (65.0±15.6 years vs. 66.2±15.3 years, p=0.004; 69.3±11.2 vs. 71.0±10.5, p=0.003; respectively)